# EMVCC: Enhanced Multi-View Contrastive Clustering for Hyperspectral Images

Fulin Luo
luoflyn@163.com
College of Computer Science,
Chongqing University
Chongqing, China

Yi Liu
liuyi@stu.cqu.edu.cn
College of Computer Science,
Chongqing University
Chongqing, China

Xiuwen Gong
gongxiuwen@gmail.com
Faculty of Engineering and IT,
University of Technology Sydney
Sydney, Australia

Zhixiong Nan
nanzx@cqu.edu.cn
College of Computer Science,
Chongqing University
Chongqing, China

Tan Guo*
guot@cqupt.edu.cn
Chongqing University of Posts and
Telecommunications
Chongqing, China

## ABSTRACT

Cross-view consensus representation plays a critical role in hyperspectral images (HSIs) clustering. Recent multi-view contrastive cluster methods utilize contrastive loss to extract contextual consensus representation. However, these methods have a fatal flaw: contrastive learning may treat similar heterogeneous views as positive sample pairs and dissimilar homogeneous views as negative sample pairs. At the same time, the data representation via self-supervised contrastive loss is not specifically designed for clustering. Thus, to tackle this challenge, we propose a novel multi-view clustering method, i.e., Enhanced Multi-View Contrastive Clustering (EMVCC). First, the spatial multi-view is designed to learn the diverse features for contrastive clustering, and the globally relevant information of spectrum-view is extracted by Transformer, enhancing the spatial multi-view differences between neighboring samples. Then, a joint self-supervised loss is designed to constrain the consensus representation from different perspectives to efficiently avoid false negative pairs. Specifically, to preserve the diversity of multi-view information, the features are enhanced by using probabilistic contrastive loss, and the data is projected into a semantic representation space, ensuring that the similar samples in this space are closer in distance. Finally, we design a novel clustering loss that aligns the view feature representation with high confidence pseudo-labels for promoting the network to learn cluster-friendly features. In the training process, the joint self-supervised loss is used to optimize the cross-view features. Abundant experiment studies on numerous benchmarks verify the superiority of EMVCC in comparison to some state-of-the-art clustering methods. The codes are available at https://github.com/YiLiu1999/EMVCC.

*Corresponding author

## CCS CONCEPTS

• **Theory of computation** → **Unsupervised learning and clustering**.

## KEYWORDS

Multi-view clustering, contrastive learning, hyperspectral images (HSIs), self-supervised learning

**ACM Reference Format:**
Fulin Luo, Yi Liu, Xiuwen Gong, Zhixiong Nan, and Tan Guo. 2024. EMVCC: Enhanced Multi-View Contrastive Clustering for Hyperspectral Images. In *Proceedings of the 32nd ACM International Conference on Multimedia (MM '24), October 28-November 1, 2024, Melbourne, VIC, AustraliaProceedings of the 32nd ACM International Conference on Multimedia (MM'24), October 28-November 1, 2024, Melbourne, Australia.* ACM, New York, NY, USA, 9 pages. https://doi.org/3664647.3681600

## 1 INTRODUCTION

Unlike traditional grayscale or RGB images, hyperspectral image (HSI) is imaged with one spectral dimension and two spatial dimensions. The spatial dimensions can provide the spatial characteristics of scenes, such as structure and texture. In particular, the spectral dimension captures fine spectral characteristics of scenes with tens to hundreds of continuous spectral bands and reveals the unique spectral fingerprint. These distinctive characteristics have enabled the wide application of HSI in the fields of mineral exploration [10], vegetation monitoring [49], and military reconnaissance [40]. High-precision interpretation of HSI always relies on abundant high-quality labeled data [12, 14, 25]. However, in practice, large amounts of data labeling are often laborious, expensive, and impractical [8, 9]. Unsupervised learning methods, especially data clustering, have received widespread attention due to their unique value in exploring the intrinsic structure and hidden patterns of unlabeled data.

In particular, HSI clustering methods distinguish and label pixels based on their inherent similarity, maximizing intra-class and minimizing inter-class similarity [4]. As a powerful unsupervised machine learning paradigm, self-supervised learning aims to learn a representation which is beneficial to various downstream tasks [29, 30, 41] without any prior knowledge. The mainstream of self-supervised learning is the contrastive learning with dual-branch,

and the self-supervised contrastive learning generates data representations by treating augmented views of data as positive samples. These models are updated by encouraging the positive samples close to each other and the negative samples to move away from each other. For example, SimCLR [5] learns feature representation by maximizing the consistency between different augmented views of the same data samples. BYOL [11] relies on interaction and mutual learning online. MoCo [17] facilitates unsupervised contrastive learning by constructing a dynamic dictionary with queues. SwAV [3] enforces consistency between cluster assignments from different views of the same image. Contrastive learning has also been widely used in HSI clustering [13, 15]. However, although these methods have achieved good results, the collision problem [1] of contrastive learning remains unsolved. Meanwhile, a single neighbor pixel of HSI can make this problem more prominent due to extremely similar spatial features.

To address this challenge in contrastive learning, some studies [24, 28, 34, 39, 45] proposed to combine multi-view clustering with contrastive learning by integrating diverse and complementary information between different views. CMSCGC [15] proposed a novel deep multi-view subspace clustering algorithm to learn textural and spectral-spatial information. SSMLC [37] proposed a correlation coefficient-based spectral delineation method to generate multiple spectrum-views to obtain a consistent representation of HSI. GoMIC [7] proposed a self-supervised multi-view image clustering technique under contrastive heterogeneous graph learning to reveal intra-class and inter-class relationships in data. [20] designed a two-branch dense spectrum-spatial network for HSI clustering to extract spectral and spatial features separately. However, most existing methods tend to focus on the consistency of the same samples in different views, and the potential for similar but distinct views in the cross feature domain is often overlooked [50].

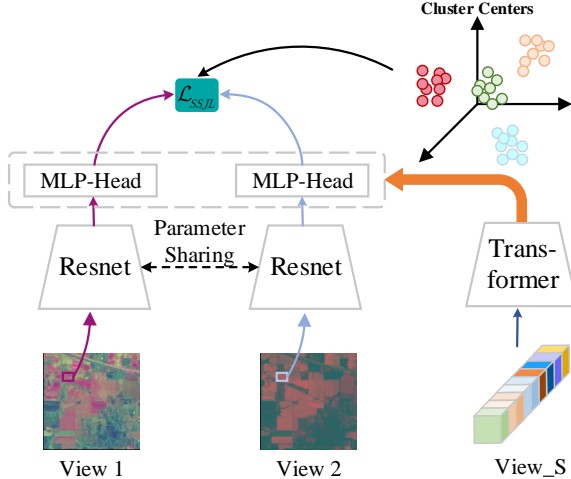

Figure 1: Overview of the proposed Enhanced Multi-View Contrastive Clustering (EMVCC). In EMVCC, the spatial-view (View 1 and View 2) is enhanced using spectrum-view (View_S) features to increase the similarity between positive pairs and the difference between negative pairs.

In this study, we propose a novel Enhanced Multi-view Contrastive Clustering (EMVCC), as illustrated in Figure 1, for extracting cross-view consensus representation of HSI. Unlike natural images, HSIs have rich spectral information. In natural images, multi-views are obtained by data augmentation methods. In EMVCC, we use the spectral segmentation method to obtain multi-view spatial features. Different from existing multi-view constrative clustering methods that exploit the coherence goal of latent features to explore common semantics of all views. Considering the high similarity of information of neighbor heterogeneous spatial-multi-view, it is difficult for general contrastive models to regard them as negative sample pairs. Therefore, EMVCC utilizes Transformer [43] to extract the global correlation information from the spectrum-view to enhance the difference of spatial-multi-view of neighbor samples. This provides a novel paradigm for follow-up work in using cross-scene view features to reduce the homogeneity between single-scene views. After that, EMVCC optimizes the extracted consensus representation using self-supervised joint loss, which efficiently explores the common semantics in all views to obtain a discriminative "cluster-friendly" consensus representation. Compared with existing clustering methods, extensive experiments on four HSI datasets have demonstrated the effectiveness and superiority of our proposed EMVCC.

The main contributions of this study can be summarized as follows.

- We propose a novel Enhanced Multi-view Contrastive Clustering (EMVCC), which utilizes corss feature domain views to reduce homogeneity between single scene views to extract a cross-view consensus representation of HSI.

- We design a Self-supervised Joint Loss to efficiently explore the common semantics in all views to obtain distinctive clustering-friendly consensus representation.

- We allocate the samples based on the similarity between attribute features and cluster centers, allowing the network to output data labels directly without post-processing.

- We conduct comprehensive experiments on HSI to validate the effectiveness of Self-supervised Joint Loss and the proposed network.

## 2 RELATED WORK

### 2.1 Contrastive Learning

As one of the most effective self-supervised methods, contrastive learning aims to learn data representation by comparing the similarity between positive and negative samples. With the principle of pulling positive pairs close and promoting negative pairs far, contrastive learning methods, such as InfoNCE [33], SimCLR [5], MoCo [17], BYOL [11], SwAV [3] have been widely studied in computer vision.

In Multi-view clustering (MVC) [15, 16, 50], the idea of contrastive learning is utilized to compare the similarity and difference between different views to improve data representation and cluster performance. CMC [38] found that the more views are introduced, the better the generated representation can capture the semantic information of the underlying scene information. DealMVC [50]

proposed a dual contrastive calibration mechanism to keep the consistency of similar samples while to make full use of the diversity of multi-view information for different samples in cross-scene view. MFLVC [45] introduced two objectives for the multi-view clustering of high-level features and the generation of pseudo labels, employing contrastive learning.

## 2.2 Multi-view Clustering

MVC has attracted increasing attention due to its excellent performance in the unsupervised domain in recent years. MVC completes the knowledge extraction task by fusing complementary and consensus information from each view. Existing multi-view clustering approaches mainly include four types: graph-based MVC, multiple kernel-based MVC, subspace-based MVC, and scalable-based MVC.

The graph-based MVC [22, 23] utilizes graph topology and connectivity between nodes to capture the intrinsic feature and the associative information of the data to improve the clustering performance. [32] proposed an auto-weighted multiple graph learning (AMGL) framework to learn a set of weights automatically for all the graphs. The multiple kernel-based MVC [6] is able to capture the nonlinear structure inherent in many real-world data. [27] proposed a multiple kernel k-means (MKKM) clustering with a matrix-induced regularization to reduce redundancy and enhance the diversity of the selected kernels. The subspace-based MVC [36, 44] assumes that data are usually located in a subspace of low dimension embedding. [31] designed a subspace representation learning method that simultaneously learned a view-consistent representation and a set of view-specific representation for multi-view subspace clustering. [52] proposed a latent multi-view subspace clustering (LMSC) method, which clustered data points with latent representation. The ensemble-based MVC integrates clustering results from different views to obtain the final clustering results. The scalable-based MVC is a clustering method for processing large-scale data sets. [21] proposed a scalable and parameter-free graph fusion framework for multi-view clustering. [54] proposed a scalable multiplex network embedding model to represent information on multi-type relations in a unified embedding space.

The above methods can fully utilize the information of multiple views, but they do not capture shallow representation of multi-view data, and the discriminative ability of the obtained representation is thus limited. Benefiting from robust feature representation, deep networks [19, 45, 51, 53] have the ability to extract finer feature representation. Through deep multi-view clustering, potential useful consensus representation of multi-view data can be efficiently discovered for improving data clustering performance.

## 3 METHOD

This section proposes a novel enhanced multi-view contrastive clustering, termed EMVCC. The framework is shown in Figure 2.

## 3.1 Multi-view Construction

For multi-view data, it is significant to make good use of the information from different views. Obviously, different bands in the HSI can be considered as different views because they can reflect different properties of the same scene. To construct multi-view data, we use spectral segmentation to construct multi-view by treating spectral features as different views, and we use Principal Component Analysis (PCA) [35] to reduce the dimensionality of the divided data. Subsequently, a sliding window is used to extract the pixel and its adjacent neighbor pixels, and the data points are represented as patches. Finally, the patches are transformed by using multiple data augmentation methods (such as cropping, flipping, rotating, and shearing) to generate multi-view features along the spatial dimension. Therefore, we can obtain multi-view features $\{X^v\}_{v=1}^{V}$, where $X^p \in R^{N \times p \times p \times d}$, $p$ is the patch size, $d$ is the dimension of the spatial-view. In other branch, we use the spectral information of the pixels as additional spectrum-view data $S \in R^{N \times d}$, $d$ is the dimension of the spectrum-view.

## 3.2 Spectrum-Enhanced Spatial View Features

To fully utilize the spatial spectral information of HSI, we designed a two-branch deep neural network. The feature extraction module maps multi-view samples to feature vectors.

In the spatial model, we use ResNet [18] as the basic feature extractor. In the $\{X^v\}_{v=1}^{V}$ spatial-view, the spatial information of each location is included. By ResNet, the complex spatial features in the views can be effectively captured. We make $x_i^1$ and $x_i^2$ denote the first view and second view of spatial-view respectively. In fact, the spatial model is equivalent to a nonlinear embedding function $f_R(\cdot)$, which maps the inputs $x_i^1$ and $x_i^2$ to deep features $hR_i^1$ and $hR_i^2$. The formula is as follows.

$$hR_i^v = f_R(x_i^v) = ResNet(x_i^v) \tag{1}$$

where $hR_i^v \in R^{N \times b}$ is the output after the average pooling layer. Then, we use a MLP with one hidden layer as the projection head $proj(\cdot)$ and the classification head $pred(\cdot)$ respectively, and map the representation $hR_i^v$ into the contrastive loss space and classification space respectively, so as to obtain $zR_i^v = proj(hR_i^v) = \sigma(W_P \cdot hR_i^v + b_P)$ and $cR_i^v = pred(hR_i^v) = \sigma(W_C \cdot hR_i + b_C)$, where $\sigma$ is the ReLU nonlinearity and the $W$ and $b$ are the weight and bias parameters.

In the spectrum model, considering that the spectral data of each pixel from the HSI changes continuously, the global relationship is crucial to understand the entire spectrum-view. Therefore, we use Transformer [43] to extract global relevant information of the spectrum-view. We use $S_i$ to represent the samples in the spectral dimension. In the standard Transformer architecture, the architecture consists of multi-head self-attention and MLP blocks. Furthermore, a layer-norm operation is employed before each block and residual connections are merged after each block. Transformer applies a self-attention layer to model the global relationship between the input embeddings, which improves the global modeling ability of the spectral information in the HSI data. The formula is as follows.

$$hT_i = f_T(S_i) = Transformer(S_i) \tag{2}$$

For each pixel in HSI, adjacent pixels of different spectral channels are strongly correlated, and adjacent pixels often belong to the same object (especially for very high-resolution data). However, adjacent pixels that are not in the same category also have similar spatial characteristics. General contrastive loss directly utilizes the data representation obtained by these spatial features, and these

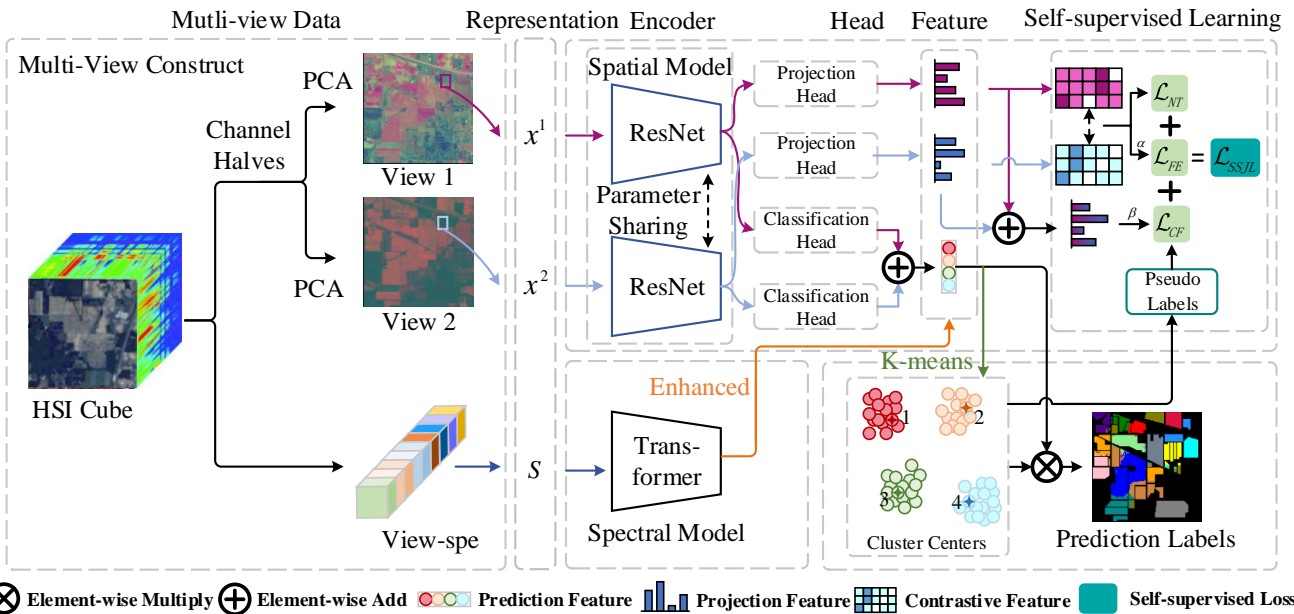

**Figure 2: Overall framework of the proposed Enhanced Multi-View Contrastive Clustering (EMVCC) which consists of three key components: Multi-view Construction, Spectrum-Enhanced Spatial View Features, and Self-supervised Joint Loss. The Spectrum-Enhanced Spatial View Feature module maps the multi-view data from the inputs as feature vectors. Subsequently, the Self-supervised Joint Loss is devised to mine potential clustering-friendly features in the multi-view space.**

samples will be regarded as false negative pairs. In order to enhance the similarity between the positive samples and maximize the difference between the negative samples, we used the extracted data representation $hT_i$ from the spectral module to strengthen the extracted data representation $zR_i^v$ from the spatial model, and the subsequent experiments have demonstrated the effectiveness of this approach. The consensus representation $z_i^v$ is expressed as follows.

$$z_i^v = zR_i^v + proj(hT_i) \quad (3)$$

where $proj(\cdot)$ denotes the projection head used after the Transformer block. In order to retain sufficient categorization information in the classification space, we fuse the label prediction probabilities of multi-view as follows.

$$c_i = c_i^1 + c_i^2 + pred(hT_i) \quad (4)$$

where $pred(\cdot)$ denotes the classification head used after the Transformer block.

## 3.3 Self-supervised Joint Loss

For data clustering, we employ the self-supervised loss to effectively integrate both the spatial model and the spectrum model for joint training. Our overall loss function is defined as follows:

$$\mathcal{L} = \mathcal{L}_{NT} + \alpha\mathcal{L}_{FE} + \beta\mathcal{L}_{cf} \quad (5)$$

where $\alpha \geq 0$ and $\beta \geq 0$ are two hyperparameters to balance the contribution between $\mathcal{L}_{FE}$ and $\mathcal{L}_{cf}$.

*3.3.1 Contrastive Loss.* Contrastive learning can learn representation from unlabeled data. It can sample positive pairs, which should be similar samples with similar representation. We use SimCLR [5], which learns representation by maximizing consistency between different augmented views of the same data samples with contrastive loss. More specifically, the positive pair in our network is usually obtained by taking two views of the same sample and then yielding $z_{2k-1}$ and $z_{2k}$ for contrastive loss, the contrastive loss $\mathcal{L}_{NT}$ between a pair of positive cases $i$ and $j$ is as follows:

$$\ell_{i,j}^{NT-Xent} = -log\frac{exp(sim(z_i^1, z_j^2)/\tau)}{\sum_{k=1}^{2N} 1_{k \neq i}exp(sim(z_i^1, z_k^2)/\tau)} \quad (6)$$

where $sim(\cdot, \cdot)$ is cosine similarity between two vectors, and $\tau$ is a temperature scalar.

*3.3.2 Feature Enhanced Loss.* We apply multi-view consistent contrastive learning for the consistency representation to distinguish samples more clearly in the feature space. Based on this, we adopt and improve the probabilistic contrast loss function of DCEnet [30], which can stabilize the model and find the boundaries of different classes by constraining the output of the siamese branch based on cosine similarity.

$$\mathcal{L}_{FE} = -mean(D(z^1, StopGrad(z^2)) + 1 \quad (7)$$

where $StopGrad(\cdot)$ is a common technique used to implement parameter sharing between branches and facilitate successful contrastive learning. This technique typically involves stopping the gradient propagation in one branch while only receiving gradient

information from the other branch when computing the similarity between the outputs of the two branches. $D(\cdot, \cdot)$ represents the calculation of cosine similarity, which can be defined as follows.

$$D(z^1, StopGrad(z^2)) = \frac{z^1 \cdot StopGrad(z^2)}{\|z^1\|_2 \cdot \|StopGrad(z^2)\|_2} \quad (8)$$

where $\|\cdot\|_2$ is $\ell_2$-norm.

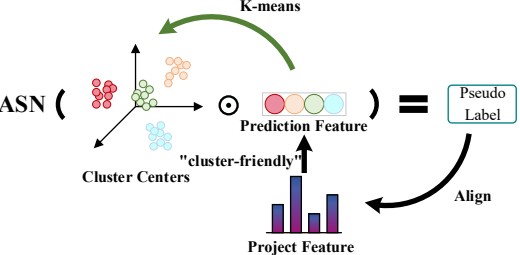

**Figure 3: Cluster-Friendly Loss. It aligns multi-view feature representation and high-confidence pseudo-labels to constrain network learning "clustering-friendly" features.**

*3.3.3 Cluster-Friendly Loss.* Since the data representation obtained by contrastive learning are not specifically designed for clustering, the use of these representation features directly makes the clustering effect unsatisfactory. For this reason, we design a novel clustering loss as in Figure 3 that aligns consensus representation of view features with high-confidence pseudo-labels to learn clustering-friendly representation that captures similarity between instances. Its expression is as follows.

$$\mathcal{L}_{cf} = CE(z_{proj_i}, ASN(z_{pred_i} \times \mu_i^T))) \quad (9)$$

where $CE$ denotes the use of a cross-entropy loss function, $\mu_i$ is $i$th cluster center, and $ASN$ operation is used to preprocess the output of model to make it more suitable for subsequent clustering tasks. More specifically, we first use *Normalize* to make the distribution of the representation smoother, helping the model learn more discriminative and semantically informative representation. Then, *Softmax* is used to convert the raw scores into probability distribution to reduce the differences between the representation. Finally, *Argmax* is used to convert the model's output into pseudo-labels.

## 4 EXPERIMENTS

In this section, four benchmark datasets are used to analyze our EMVCC. Subsequently, the experimental results are elaborated in depth through a detailed analysis.

## 4.1 Experimental Setup

*4.1.1 Benchmarks.* Since HSIs have too many bands and are difficult to process, the general approach uses dimensionality reduction or band selection to reduce the number of bands for clustering. However, it will cause information loss, because different bands contain different land-cover information. Therefore, we simply split spectral channels in half to create multi-view data, preserving key spectral information and providing diverse views to achieve the contrast of data. In experiments, we adopt the same multi-view

**Table 1: Information about the four HSI datasets.**

| Dataset | Clusters | Samples | Views | Patch_size | Bands |
|---------|----------|---------|-------|-----------|-------|
| Salinas | 16 | 54129 | 3 | 28×28×3 | 224 |
| Botswana | 14 | 3248 | 3 | 28×28×3 | 145 |
| Indian Pines | 16 | 10249 | 3 | 28×28×3 | 200 |
| Houston | 15 | 15029 | 3 | 28×28×3 | 144 |

data input for the other methods. To ascertain the efficacy of our EMVCC, we conducted comprehensive experiments on the four HSI datasets: Salinas, Botswana, Indian_Pines, and Houston. Table 1 provides a brief overview of these datasets. To quantify performance, we utilize a serial of clustering performance metrics, including overall accuracy (OA), Fowlkes-Mallows index (FMI), adjusted rand index (ARI), adjusted mutual information (AMI), and normalized mutual information (NMI), as well as employing purity and kappa coefficient (Kappa) to evaluate the clustering performance on the datasets. Across all these methods, the number of clusters is set by referencing the ground truth in each dataset. Subsequently, we evaluated clustering validity of EMVCC in comparison to several state-of-the-art (SOTA) models, including MFLVC (CVPR 2022) [46], GCFAgg (CVPR 2023) [47], ACCMVC (TNNLS 2024) [48], CMSCGC (TGRS 2024) [15], GCOT (TGRS 2022) [26], EKGCSC (TGRS 2020) [2], DSCRLE (TGRS 2023) [55] and SDST (TGRS 2024)[29].

*4.1.2 Implementation Details.* EMVCC was implemented in Python (version 3.10.6) using the PyTorch framework. We use Transformer and ResNet-50 as the backbone for the spectral model and the spatial model, respectively. And the number of heads for the Transformer was set to 6. The classification head ($pred(\cdot)$) has one layer MLP, and its output $c$ does not have BN or ReLU. The hidden $c$ is 2048. This MLP has one layer MLP. The projection Head ($proj(\cdot)$) also has one layer MLP with ReLU and the dimensions of input and output are 2048 and 128. In the self-supervised joint loss, the hyperparameters $\alpha$ and $\beta$ were set to 1 and 0.1. The model was trained by Adam optimizer with an initial learning rate of 0.0001, and the weight decay is 0.000001. The computational environment was conducted on a computer with an NVIDIA RTX A6000 48G GPU.

## 4.2 Experimental Results and Analysis

*4.2.1 Quantitative Evaluation.* We compare EMVCC with six baselines, including traditional subspace clustering methods (GCOT [26], EKGCSC[2]), self-supervised deep clustering methods (DSCRLE [55], SDST [29]) and contrasting multi-view clustering methods (MFLVC [46], GCFAgg [47], ACCMVC [48] and CMSCGC[15]). Table 2 lists the clustering performance of all compared methods on the four datasets. From these results, we make the following observations. Compared with these clustering algorithms, our EMVCC consistently achieves the most favorable clustering results in most datasets. For example, when examining Salinas' results, EMVCC outperforms the second best method by 1.29%, 1.95%, 9.13%, 2.83%, 2.22%, 2.90%, and 2.22% in OA, Kappa, Purity, ARI, AMI, FMI, and NMI, respectively. We speculate the reason is that these self-supervised methods do not consider the multi-view information of data. Compared with deep multi-view clustering methods, our

**Table 2: Clustering performance on the four HSI datasets. The most outstanding results are denoted in bold, while the second-best values are underlined. The notation " − " signifies an out-of-memory error encountered during the training process.**

| Methods | Salinas | | | | | | | Botswana | | | | | | |
|---|---|---|---|---|---|---|---|---|---|---|---|---|---|---|
| Metrics (%) | OA | Kappa | Purity | ARI | AMI | FMI | NMI | OA | Kappa | Purity | ARI | AMI | FMI | NMI |
| MFLVC[46] | 74.20 | 71.80 | 78.26 | 57.40 | 75.61 | 61.71 | 75.63 | 70.67 | 68.39 | 73.40 | 55.99 | 69.21 | 59.34 | 69.54 |
| GCFAgg[47] | 50.13 | 45.24 | 56.04 | 37.29 | 53.14 | 43.29 | 53.18 | 58.53 | 55.34 | 62.11 | 40.45 | 54.84 | 44.95 | 55.33 |
| ACCMVC[48] | 73.41 | 71.09 | 82.32 | 60.10 | 76.76 | 64.35 | 76.78 | 70.47 | 68.16 | 73.50 | 56.94 | 68.62 | 60.21 | 68.94 |
| CMSCGC[15] | − | − | − | − | − | − | − | 86.98 | 85.46 | 86.98 | 75.14 | 85.31 | 77.12 | 85.89 |
| GCOT[26] | 68.48 | 66.00 | 73.90 | 61.33 | 81.34 | 68.48 | 81.36 | 61.76 | 58.24 | 62.41 | 60.99 | 80.73 | 67.94 | 80.94 |
| EKGCSC[2] | 62.12 | 57.59 | 64.28 | 57.40 | 74.12 | 63.65 | 74.14 | 65.06 | 61.82 | 66.26 | 56.59 | 82.96 | 65.23 | 83.15 |
| DSCRLE[55] | 75.25 | 72.68 | 77.83 | 62.10 | 81.56 | 65.93 | 81.57 | 92.73 | 92.14 | 92.73 | 88.26 | 92.43 | 89.15 | 92.51 |
| SDST[29] | 75.80 | 73.12 | 76.09 | 60.50 | 78.37 | 64.56 | 78.39 | 74.26 | 72.09 | 75.03 | 62.93 | 76.20 | 66.23 | 76.45 |
| EMVCC | **77.09** | **75.07** | **86.96** | **64.93** | **83.78** | **68.83** | **83.79** | **94.40** | **93.39** | **94.40** | **88.51** | **94.28** | **89.41** | **94.34** |
| Methods | Indian_Pines | | | | | | | Houston | | | | | | |
| Metrics (%) | OA | Kappa | Purity | ARI | AMI | FMI | NMI | OA | Kappa | Purity | ARI | AMI | FMI | NMI |
| MFLVC[46] | 53.17 | 49.32 | 64.73 | 36.52 | 56.28 | 43.81 | 56.28 | 42.08 | 37.58 | 44.71 | 23.44 | 39.40 | 29.16 | 39.55 |
| GCFAgg[47] | 38.14 | 32.13 | 50.27 | 22.05 | 37.05 | 30.24 | 37.33 | 26.19 | 20.67 | 28.53 | 11.34 | 20.30 | 17.94 | 20.50 |
| ACCMVC[48] | 55.80 | 52.43 | 70.51 | 37.84 | 61.28 | 45.25 | 61.45 | 45.69 | 41.51 | 49.31 | 26.59 | 41.28 | 31.99 | 41.43 |
| CMSCGC[15] | 45.53 | 52.09 | 58.34 | 27.90 | 51.86 | 35.70 | 40.46 | 57.60 | 60.51 | 58.07 | 39.16 | 60.41 | 44.19 | 54.34 |
| GCOT[26] | 50.20 | 42.37 | 54.67 | 33.18 | 51.46 | 43.58 | 51.71 | − | − | − | − | − | − | − |
| EKGCSC[2] | 61.24 | 56.10 | 69.89 | 49.69 | 68.35 | 56.10 | 68.50 | − | − | − | − | − | − | − |
| DSCRLE[55] | 48.88 | 43.50 | 56.83 | 29.85 | 50.90 | 37.61 | 51.13 | 61.29 | 58.32 | 62.53 | 43.61 | 62.69 | 48.11 | 62.79 |
| SDST[29] | 54.22 | 47.49 | 57.62 | 34.63 | 51.44 | 43.45 | 51.68 | 54.93 | 51.33 | 55.15 | 33.18 | 53.59 | 38.93 | 53.71 |
| EMVCC | **65.13** | **61.84** | **77.08** | **51.93** | **69.67** | **58.02** | **69.81** | **70.36** | **68.14** | **72.91** | **57.14** | **72.72** | **60.34** | **72.79** |

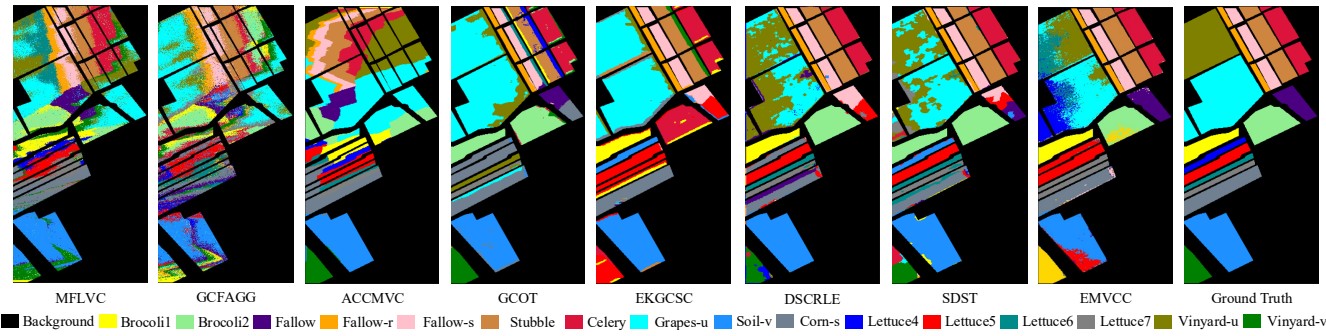

**Figure 4: Clustering maps of Indian_Pines.**

**Figure 5: Clustering maps of Salinas.**

method also has obvious advantages, which also verifies that single-view data is difficult to provide sufficient information for clustering. In addition, the occurrence of false negative pairs also affects the

clustering performance. In summary, the above observations confirm the superior performance of our proposed EMVCC.

*4.2.2 Qualitative Evaluation.* Figures 4-5 show the visual classification maps of different methods on the Indian_Pines and Salinas

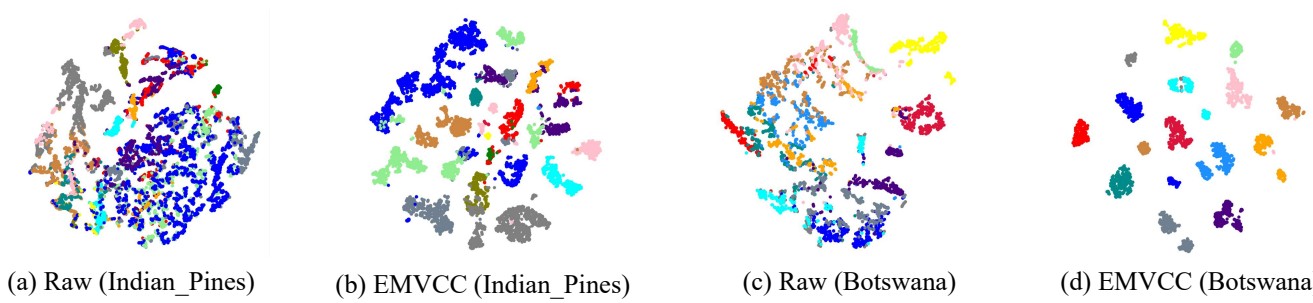

(a) Raw (Indian_Pines)  (b) EMVCC (Indian_Pines)  (c) Raw (Botswana)  (d) EMVCC (Botswana)

**Figure 6: Visualization of t-SNE on the Indian_Pines and Botswana datasets.**

datasets. As can be seen from the plots, the regions acquired by our proposed EMVCC are more homogeneous and have the least number of misclassified pixels compared with all the other benchmark methods. This is especially evident in the regions where the mixed ground objects have similar spectral characteristics. Taking the Indian_Pines dataset as an example (see Figure 4), the pixels of "Soybean-mintill" and "Soybean-notill" land cover types are prone to be misclassified by most of methods due to their overly similar spectral features. Most notably, our method effectively distinguishes between these two objects. Our method employs an advanced multi-view feature extraction method that effectively identifies and exploits relevant features in HSI data. By comparing the clustering plots, we show the difference in identifying the feature continuity of our method on the Indian Pine dataset. The clustering plots clearly show that our method has a significant advantage in terms of edge sharpness and continuity on the Indian_Pines as well as Salinas datasets. The classification plot shows that the proposed EMVCC significantly improves the discrimination of similar ground objects. In addition, we visualized the learned embedding distribution of EMVCC on the Indian_Pines and Botswana datasets by the t-SNE [42]. From the results in Figure 6, we conclude that EMVCC is able to reveal the intrinsic clustering structure better than the original features.

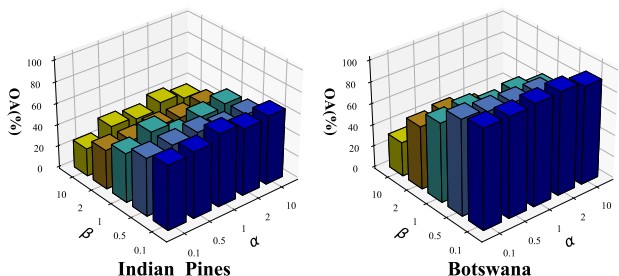

**Figure 7: Analysis of parameters $\alpha$ and $\beta$ in the overall loss function.**

## 4.3 Parameter Analysis

In EMVCC, the overall loss function primarily has two parameters: $\alpha$ and $\beta$, which have a significant impact on the clustering performance. We use a grid search strategy to find the optimal $\alpha$ and $\beta$

on the Indian_Pines and Botswana datasets. As can be seen from Figure 7, the best clustering results are obtained when $\alpha$ and $\beta$ are set to 1 and 0.1 respectively. Thus, by setting $\alpha$ and $\beta$ to 1 and 0.1, our method achieves satisfactory performance.

**Table 3: Ablation study on the Indian_Pines datasets (%)**

| DAtastes | EMVCC-1 | EMVCC-2 | EMVCC-3 | EMVCC-4 | EMVCC-5 | EMVCC-6 |
|---|---|---|---|---|---|---|
| Spatial model | ✓ | × | ✓ | ✓ | ✓ | ✓ |
| Spectrum model | × | ✓ | ✓ | ✓ | ✓ | ✓ |
| Feature Enhanced Loss | ✓ | ✓ | × | ✓ | × | ✓ |
| Cluster-Friendly Loss | ✓ | ✓ | × | × | ✓ | ✓ |
| Kappa | 54.03 | 27.06 | 56.52 | 58.60 | 59.87 | 61.84 |
| AMI | 59.60 | 27.09 | 65.61 | 68.53 | 64.98 | 69.67 |
| OA | 57.76 | 35.43 | 60.04 | 61.91 | 63.26 | 65.13 |

## 4.4 Ablation Study

In this experiment, the impact of Spatial model, Spectrum model, Feature Enhanced Loss and Cluster-Friendly Loss of EMVCC is evaluated on the Indian_Pines dataset. The results are listed in Table 3. These methods are simplified from EMVCC-1 to EMVCC-6. The results reveal the necessity of each key component in EMVCC. Specifically, the clustering accuracy of EMVCC-6 is higher than that of EMVCC-1 and EMVCC-2, indicating that the spectrum-enhanced spatial-view feature is effective. Compared with EMVCC-3, the introduction of Feature Enhanced Loss enhances the consensus representation in EMVCC-4 producing better results for EMVCC, which confirms the important role of Feature Enhanced Loss. Similarly, the accuracy of EMVCC-5 is significantly better than that of EMVCC-3, demonstrating that the Cluster-Friendly Loss enables EMVCC to produce a feature representation that is more favourable to clustering. Finally, the OA value of EMVCC-6 exceeds that of EMVCC-1 to EMVCC-5, proving the superiority of EMVCC. Thus, the proposed Spectrum-Enhanced Spatial View Feature, Feature Enhanced Loss and Cluster-Friendly Loss both improve the clustering accuracy of EMVCC.

## 5 CONCLUSION

In this paper, we design a novel Enhanced Multi-View Contrastive Clustering, called EMVCC. Specifically, we propose an enhanced-view mechanism to increase the differentiation between dissimilar samples and the similarity between similar samples. Then, we designed a Self-supervised Joint Loss to constrain the consensus representation of different views for avoiding false negative pairs.

To further leverage the consensus representation of multi-view data, we use a contrast loss to maximize the consistency between different augmented views of the same data samples. Using Feature Enhanced Loss enhances the discriminative features to ensure the diversity of multi-view information. Finally, we also design a novel Cluster-Friendly Loss that aligns view feature representations with high-confidence pseudo-labels to constrain the network for achieving favorable clustering. Complete experiments on four HSI datasets convincingly demonstrate the effectiveness of our proposed EMVCC.

## ACKNOWLEDGMENTS

This work was supported by the National Natural Science Foundation of China under Grant 62371076, 62071340 and 62201109, and the Natural Science Foundation of Chongqing under Grant CSTB2022NSCQ-MSX0452.

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
