# OpenReview forum: "EMVCC: Enhanced Multi-View Contrastive Clustering for Hyperspectral Images"
_acmmm.org/ACMMM/2024/Conference — MM2024 Poster_

### Official Review · Reviewer_AV8W · 2024-05-21

**Rating:** 5
**Confidence:** 4

**Summary:**

The authors proposed a novel enhanced multi-view contrastive clustering for Hyperspectral Images(HSIs), namely EMVCC, to mitigate the 'collision problem' in contrast learning as well as to obtain clustering-friendly consensus representations. It utilizes Transformer to extract the global correlation information of spectrum-view to enhance the difference of spatial-multi-view of neighbor samples. This provides a new paradigm for subsequent work that utilises cross-scene view features to reduce homogeneity between single scene views. A self-supervised joint loss to efficiently explore the common semantics in all views to obtain distinctive clustering-friendly consensus representations. Experiments have shown better clustering performances compared to Multi-view Clustering methods. The model allocate the samples based on the similarity between attribute features and cluster centers, allowing the network to output data labels directly without post-processing.

**Strengths:**

1/ The Enhanced Multi-View Contrast Clustering (EMVCC) method has been clearly explained. It combines multi-view clustering and contrast learning to extract cluster-friendly consensus representations. Enhance the difference of spatial-multi-view of neighbour samples by learned features from the spectral view.This greatly alleviates the problem of false negatives due to similar heterogeneous samples in contrast learning. Simultaneously, EMVCC employs self-supervised joint loss optimization to refine the extracted consensus representations. It effectively explores the common semantics across all views to obtain differentiated, 'cluster-friendly' consensus representations.It may tackle the actual unsupervised problems related to various hyperspectral in satellite data.

2/ The presented results show that EMVCC outperforms other competing methods on multiple datasets based on clustering accuracy as well as a range of clustering performance metrics. The study of ablation is rigorously conducted, showing the impact of the spatial module, the spectral module on multi-view clustering, as well as the impact of feature enhancement loss and clustering optimisation loss in self-supervised joint loss.

3/ Visualization of clustering maps and t-SNE conducted are appreciated.

4/ This work could have a significant impact since large volume of remote sensing data are publicly available without labels. There is an urgent need of efficient and powerful unsupervised training models for many remote sensing applications, including precision agriculture and mineral exploration understanding and monitoring.

**Limitations:**

The authors have neither detailed the limits, nor the potential negative societal impacts (military or surveillance applications) of the proposed method. No negative societal impacts are expected.

#### question

1/ Figure 2: 'Anchor Centers' blocks are not defined.

2/ Typo: lack of consistency between "multiview" and "multi-view"

3/ Typo: Unstandardised use of "spectral" and "spectrum"

4/ Table 1: The first letter of ‘salinas’ is not capitalised.

5/ Typo: lack of consistency between "View_S" and "View_Spe"

**Suitability:**

3

---

### Official Review · Reviewer_7YqP · 2024-05-23

**Rating:** 4
**Confidence:** 2

**Summary:**

This paper has proposed the Enhanced Multi-View Contrastive Clustering for Hyperspectral Images. To avoid false negative pairs, the authors have designed the self-supervised joint loss to constrain the consensus representation of different views.  Experiments on several hyperspectral image datasets are conducted to investigate the performance of the proposed method.

**Strengths:**

1、	The design of the self-supervised joint loss.
2、	A clustering loss that aligns view feature representations with high-confidence pseudo-labels for constraining the network.

**Limitations:**

1、	In Abstract, the authors stated that “these methods have a fatal flaw: Contrastive learning treats similar heterogeneous views as positive sample pairs and dissimilar homogeneous views as negative sample pairs.” This sentence explains the main idea of contrastive learning without describing the so-called fatal flaw.
2、	From Abstract, I cannot understand what special designs for hyperspectral images which are significantly different from natural images.
3、	Why do not give the accuracy of all hyperspectral image clustering methods?
4、	Many mistakes in grammar, for example
(1)	the data is projected into a representation spatial with semantic information,
(2)	Abundant experiment studies on numerous benchmarks verify the superiority

**Suitability:**

2

---

### Official Review · Reviewer_HDST · 2024-05-23

**Rating:** 2
**Confidence:** 4

**Summary:**

The paper presents an Enhanced Multi-View Contrastive Clustering model for Hyperspectral Images (HSIs). The proposed model generates multi-view data from HSIs and performs constrastive learning to achieve clustering for HSIs. Experiments seem to demonstrate the effectiveness of the proposed model.

**Strengths:**

The proposed model is easy to follow and  understand.

The experiments are sufficient to verify the proposed model.

The proposed model shows competitive performance compared to other methods.

**Limitations:**

The motivation is unclear. There are no related interpretation about the difference of data representations designed for clustering or other applications that is pointed in the Abstract.

It seems that there is an error for “a representation spatial with semantic information”  at Line 28 in the Abstract.

Related review of hypersepctral image clustering is very limited and should be improved. Besides, there is also limited review for constrative learning in HSI clustering.

The citations are not correct at LIne 107 in the Introduction. They are alll designed for general  multi-view data instead of  HSI data.

In Figure 1, “space model” should be corrected as “spatial model”.

What is the anchor centers in Figure 1. There are no  related descriptions in the main text.

Why do you perform channel halves for spectral segmentation to construct multi-view data? Please give the detailed reason.

What is ASN operation? The full name of ASN is not clear.

How to achieve classification head since there are no labels in clustering tasks.

For the compared methods, [20], [23], [49] are HSI clustering methods, while others are general multi-view clustering methods. NO multi-view clustering methods designed for HSIs are compared. More related methods should be compared in the experiments, such as Tensorial multiview subspace clustering for polarimetric hyperspectral images and Contrastive Multiview Subspace Clustering of Hyperspectral Images Based on Graph Convolutional Networks.

What is the input for general multi-view clustering? Please provide detailed descriptions about the input of  general multi-view clustering methods in the experiments.

**Suitability:**

2

---

### Official Review · Reviewer_4r9Q · 2024-05-24

**Rating:** 4
**Confidence:** 2

**Summary:**

The authors propose a novel multi-view clustering method called Enhanced Multi-View Contrastive Clustering (EMVCC), which utilizes spatial multi-view learning to extract diverse features for contrastive clustering. Additionally, they employ Transformer to extract the global correlation information of spectral views to enhance differences between neighboring samples. Furthermore, they design a joint self-supervised loss to effectively avoid the generation of negative sample pairs, enhancing features with probabilistic contrastive loss to maintain diversity. Data is projected into a semantic information-rich representation space, ensuring that similar samples are closer in distance. Finally, a new clustering loss is designed to encourage the network to learn cluster-friendly features.

**Strengths:**

1.The authors propose a novel Enhanced Multi-View Contrastive Clustering (EMVCC) method, which reduces homogeneity between single-scene views using cross-feature-domain views to extract cross-view consensus representations of HSI.
2.The authors design a self-supervised joint loss to efficiently explore common semantics in all views, thereby obtaining distinctive clustering-friendly consensus representations.
3.They allocate samples based on the similarity between attribute features and cluster centers, allowing the network to directly output data labels without post-processing.

**Limitations:**

1.	The computational complexity of the model was not calculated; could the performance improvement be due to model complexity?
2.	Only six methods were compared; it is recommended to include more comparison methods.
3.	The time consumption of the model was not shown.
4.	The novelty of the research problem addressed by the method is average.
5.	The number of datasets used in the experiments is limited; it is recommended to conduct comparisons on more datasets.

**Suitability:**

2

---

### Meta-Review · Area_Chair_qwyL · 2024-06-27

**Recommendation:** Accept (Poster)
**Confidence:** 3

**Metareview:**

1 Accept, 2 Borderline Accept, 1 Reject

Most reviewers have positive comments to this manuscript except one reviewer recommends "Reject". I follow the majority side. The following issues need to be carefully addressed in the final paper.
-- unclear motivation
-- novelty of the research problem addressed by the method is average
-- more experiments and comparisons
-- Why perform channel halves for spectral segmentation to construct multi-view data
-- What is the input for general multi-view clustering
-- discuss the limits or the potential negative societal impacts of the proposed method